# Acetylcholinesterase-like proteins are a major component of reproductive trail mucus in the invasive pest land snail, *Theba pisana*

Inaliguyau R. T. Lutschini[1,2], Kate R. Ballard[1,2], Tianfang Wang[1,2], Scott F. Cummins [1,2]*

1 Centre for Bioinnovation, University of the Sunshine Coast, Maroochydore BC, Queensland, Australia,
2 School of Science, Technology and Engineering, University of the Sunshine Coast, Maroochydore BC, Queensland, Australia

* scummins@usc.edu.au

## Abstract

Invasive invertebrate pests have become a major threat to food security as global populations increase. Pesticides, often containing organophosphates, have long been used as agents for providing immediate short-term recovery, yet are often broad-spectrum, leading to the development of resistance. In insect species, one mechanism for resistance is known to be driven by mutations in acetylcholinesterase (AChE), an enzyme that catalyses the hydrolysis of acetylcholine. In this study, we explored a potential role for resistance-modified AChE in invasive pest land snails, using the Mediterranean snail *Theba pisana*. Following tissue transcriptomic investigation, an expanded family of AChE-like genes were identified that clustered phylogenetically into three individual clades, with one clade including vertebrate AChE. The majority of *T. pisana* AChE-like genes demonstrated the highest expression in the snail mucous gland during its reproductive stage. Subsequent proteomic analysis of trail mucus at the reproductive stage identified four AChE-like proteins as a major component. Immunolocalisation revealed that AChE-like protein(s) were prominent in the mucous gland secretory cells and widespread throughout the reproductive stage trail mucus, yet were largely absent from trail mucus at the non-reproductive stage. In summary, this study established a potential role for resistance-modified AChE-like proteins in pest land snail pesticide resistance via their deployment into trail mucus that may bio-scavenge organophosphates, rendering them ineffective. Their abundance during the reproductive stage is likely due to the snail's increased mobility, following periods of immobile aestivation.

## Introduction

Invasive pest land snails have become an increasing threat to human activities such as trade, agriculture and transportation. These snails have become invasive pests

**Data availability statement:** Sequence datasets were deposited in the NCBI Sequence Read Archive (SRA) database under accession number PRJNA858108.

**Funding:** The author(s) received no specific funding for this work.

**Competing interests:** The authors have declared that no competing interests exist.

due to their adaptability, rapid reproduction and lack of natural predators in their new habitats [1–3]. Pest snails are particularly noted for their agricultural damage, feeding upon a wide variety of crops, including fruits, vegetables and grains, which can lead to significant losses for farmers [2,4]. Examples of invasive pest land snails include the giant African land snail (*Achatina fulica*), Rosy Wolf Snail (*Euglandina rosea*) and Asian Tramp snail (*Bradybaena similaris*) [5–10], as well as a variety of Mediterranean land snails. These are some of the most devastating invasive snail pests.

Mediterranean land snails are naturally distributed throughout the Mediterranean regions of Europe (e.g., Spain, Italy, France, Egypt), and include the white Italian snail, *Theba pisana* (O. F. Müller, 1774) and vineyard snail, *Cernuella virgata* (da Costa, 1778) [11,12]. The earliest recording of *T. pisana* in Australia was documented in the 1920s, initially in South Australia, and they have since extended their range to Victoria, Tasmania and Western Australia [11]. Although *T. pisana* were an accidental introduction to Australia [12], they were deliberately distributed to various other locations throughout the world in small but densely populated colonies, to eradicate microscopic pathogens [13–15].

*Theba pisana* is considered an agricultural pest due to its ability to aestivate on the ears and stalks of cereal crops (e.g., wheat, barley, oat); as a result, snails can be inadvertently harvested, which can clog farming machinery and contaminate grain harvest. Aestivation is a behavioural response to high temperatures that allows the snail to avoid desiccation [16,17]. Harvested crops may be downgraded or even rejected due to snail contamination, which is a severe burden to farmers. Australian crop farmers accumulate losses of over $19 million in yield per annum [1,4,12]. Also, as an invasive snail, *T. pisana* acts as a vector (carrier) for helminth parasites that cause diseases [18] that are detrimental to livestock and human health [1,19]. With changing climates, urbanization, inflation of food prices and escalated food consumption by humans, pestiferous snails are one of many drivers that influence the vulnerability of the agricultural community [20]. Grain contamination by Mediterranean snails poses a serious threat to food security, and for this reason, population control is vital to mitigating damage by utilizing regulatory monitoring and management [21].

Pest snail control by crop growers has involved stubble management, cabling, rolling, slashing and grazing [22]. Also, since the 1990s, CSIRO Montpellier (France) have investigated biological control agents to fight pest snails, including dipteran parasitoids and predators [23–25]. Flies that have a malacophagous life stage (feed on molluscs), which include those in Sciomyzidae and Sarcophagidae families, have been released to augment control measures of pest snails in the US (Hawaii), Italy, South Africa, France, and Australia. However, due to the incompatibility between host and parasite, results were deemed unsuccessful [23,26,27]. Finally, management using organophosphates (OPs; including chlorpyrifos) and molluscicide baits have been implemented, but with liberal impact [27–31].

Organophosphates that contain active metaldehyde-based ingredients and carbamate methiocarb have been predominantly used for the treatment of crop-infested areas as a form of snail control [32]. In recent years, research regarding

resistance to chlorpyrifos identified that benthonic bivalve molluscs such as the *Scapharca inaequivalvis*, are capable of overexpressing the enzyme acetylcholinesterase (AChE) from the foot and gill tissues, suggesting that the increased content of AChE may be a consequence of OP resistance [33]. Another invasive pest to the agricultural and aquacultural environments, the golden apple snail *Pomacea canaliculata*, has been identified as having morphological changes in the head, body and digestive tract when exposed to chlorpyrifos, thus employing AChE activity as a form of resistance, as well as a bioindicator for exposure to toxicological agents [34]. This aligns with what is known for insects, whereby resistance-modified AChE has been a critical evolutionary driver for pesticide resistance [35,36].

We hypothesized that *T. pisana* secrete AChE proteins into their trail mucus, which may function to neutralize pesticides. Utilizing tissue- and reproductive stage-specific transcriptomics, and supported by proteomics, we report the presence of an expanded family of AChE-like proteins that are highly secreted into trail mucus during the reproductive stage. This finding helps to provide some explanation as to the potential molecular signalling mechanisms involved in pesticide resistance and may be utilized for the future development of more sustainable biocontrol methods.

## Materials and methods

### Animals and maintenance

*Theba pisana* were collected from Lauderdale and South Arm, Tasmania (Australia), and brought to the University of the Sunshine Coast (Sippy Downs) between April-August 2023 (reproductive stage) and January-April 2024, (non-reproductive stage). Before sample collections, individuals were identified as being at the reproductive stage based on examination of mucous glands. Furthermore, the non-reproductive stage was highlighted by no visible presence of egg clutches while in culture. Animals were maintained in a safe and clean environment with an indoor temperature of 25–29°C and a controlled relative humidity between 60–70%. Snails were fed lettuce and carrots.

### Reference de novo transcriptome assembly and protein annotation

A single *T. pisana* collected during May 2019 was euthanized and the shell was removed (total weight 0.8g, shell diameter 12mm). Snail flesh was finely sliced with a clean scalpel blade and transferred into 1.5ml microtubes. After weighing, Trizol reagent (Thermo Scientific) was added, and total RNA was isolated according to the manufacturer's instructions. Total RNA yield was determined using a Nanodrop spectrophotometer 2000c (Thermo Scientific, Waltham, USA) at 260 and 280nm. RNA was sent to Novogene (Singapore, China) for quality control and Illumina NovaSeq paired-end sequencing (150bp) at 12 Gb. Clean data (clean reads) were screened from raw sequencing reads based on (1) discard reads with adaptor contamination; (2) discard reads when uncertain nucleotides constituted more than 10% of either read (N > 10%); and (3) discard reads when low-quality nucleotides (base quality less than 20) constituted more than 50% of the read. Sequence datasets were deposited in the NCBI Bioproject (PRJ) database under accession number PRJNA858108.

High-quality reads were *de novo* assembled using SOAPdevono2 (CLC Genomics Workbench, version 10.1, Qiagen, Hilden, Germany) with parameters set as follows: seqType, fq; minimum kmer coverage = 4; minimum contig length of 100bp; group pair distance = 250. Estimation of transcript expression was performed using the *de novo* RNA-seq analysis tool on the CLC Genomic Workbench software with default parameters. A transcriptome-derived protein database was prepared using the ORF predictor (http://bioinformatics.ysu.edu/tools/OrfPredictor.html).

### Tissue- and reproduction stage-specific tissue RNA-seq analysis

The albumen glands, mucous glands, foot tissue and cerebral ganglia of *T. pisana* collected in May 2020 (reproductive) and November 2020 (non-reproductive) were carefully removed, and then immediately snap-frozen on dry ice. Each

biological replicate contained tissue from 5–10 individual snails. Total RNA was isolated with Trizol reagent according to the manufacturer's instructions. Illumina sequencing was performed on three biological replicates for each reproductive stage, as described above to a depth of at least 10 Gb for each sample. Sequence datasets were deposited in the NCBI Sequence Read Archive (SRA) database under accession number PRJNA858108. Expression levels were calculated using RSEM (RNA-seq by Expect-Maximisation) and converted to Fragments Per Kilobase of transcript per Million mapped reads (FPKM) values, resulting in lists of genes upregulated and downregulated between reproductive stages. Differentially expressed genes were classified as significant if $P < 0.05$ with at least $+/-2$ log fold-change (FC). Significantly upregulated genes were further analyzed by: (1) gene ontology using 7 databases (NT, NR, Swissprot, KOG, PFAM, GO and KEGG) with E-value thresholds from 0.01 to 1E-06. (2) tBLASTn against the *in silico* predicted secretome, to identify secreted proteins.

### Identification and characterization of acetylcholinesterase-like genes

The general protocol used to identify and characterize snail genes was as described by Ballard et al., 2021 [37,38]. AChE-like proteins were searched in the *T. pisana* reference transcriptome database using protein BLAST (BLASTp) with parameters set as E-value $\leq 10^{-3}$ and over 90% query cover. SignalP 6.0 (https://services.healthtech.dtu.dk/services/SignalP-6.0/) was used to predict signal peptides in protein sequences. The EXPASY translate SIB tool (https://web.expasy.org/translate/) enabled the identification and analysis of full-length peptide sequence reading frames, as well as pI and molecular weight. Sequences with less than 100 amino acids were not evaluated. Corresponding AChE-like gene expression datasets of *T. pisana* tissues (mucous gland, albumen gland, foot muscle and cerebral ganglia) at both reproductive and non-reproductive, were obtained, then heatmaps were generated using ClustVis (https://biit.cs.ut.ee/clustvis/).

MEGA 11 software (https://www.megasoftware.net/) was used for ClustalW alignment of all *T. pisana* AChE-like protein sequences. In addition, human (*Homos sapiens*, AAA68151) and zebrafish (*Danio rerio*; NP_571921) AChE proteins were used for comparison. A maximum likelihood estimation was conducted with a parameter of 500 replications using the Bootstrap method to generate a phylogenetic tree. This was exported to the Interactive Tree-of-Life (iTOL v6; https://itol.embl.de) to annotate a graphical figure.

### Trail mucus collection, protein isolation and preparation for mass spectrometry

Trail mucus was collected using a method previously described by Ballard et al., 2021 (39). In brief, approximately 28 *T. pisana* were rinsed in water, divided into four groups (7 each), then carefully placed into 10 cm glass Petri dishes and left to crawl for 15 min under a 20W BiPi halogen lamp at room temperature (21°C) with 70% relative humidity. Snails were removed and trail mucus was flushed with 50 ml of 0.1% trifluoroacetic acid (TFA), centrifuged and added to sterile tubes, then lyophilized in a Speedvac Concentrator (Thermo Scientific) SAVANT SC25DEXP. Lyophilized pellets were resuspended in Milli-Q water and protein concentration was determined using a Nanodrop 2000c (Thermo Scientific spectrophotometer) at A280nm. Protein (approximately 20 mg) was added to 2x Laemmli loading dye (Bio-Rad) with 2-mercaptoethanol, then fractionated by SDS-PAGE using a MINI- Protean precast Gel TGX (7.5%). Following separation, the gel was stained with Coomassie Blue and the prominent gel band was removed and processed by in-gel trypsin digestion, as described in similar invertebrates [39].

### uHPLC tandem QTOF MS/MS analyses and protein identification

Trypsinized peptides were resuspended in 100 µL 0.5% TFA in Milli-Q water and analyzed by LC-MS/MS on an ExionLC liquid chromatography system (AB SCIEX, Concord, Canada) coupled to a QTOF X500R mass spectrometer (AB SCIEX, Concord, Canada) equipped with an electrospray ion source. Twenty microlitres of each sample was injected onto a 100 mm × 1.7 µm Aeris PEPTIDE XB-C18 100 uHPLC column (Phenomenex, Sydney, Australia) equipped with a

SecurityGuard column for mass spectrometry analysis. The ion spray voltage was set to 5500 V, declustering potential (DP) 100V, curtain gas flow 30, ion source gas 1 (GS1) 40, ion source gas 2 (GS2) 50 and spray temperature at 450 °C. Linear gradients of 5–35% solvent B over 10 min at 400 µL/min flow rate, followed by a steeper gradient from 35% to 80% solvent B in 2 min and 80% to 95% solvent B in 1 min were used for peptide elution. Solvent B was held at 95% for 1 min for washing the column and returned to 5% solvent B for equilibration prior to the next sample injection. Solvent A consisted of 0.1% formic acid (aq) and solvent B contained 100% acetonitrile/0.1% formic acid (aq). The mass spectrometer acquired mass spectral data in an Information Dependant Acquisition, IDA mode. Full scan TOFMS data was acquired over the mass range 350–1400 and for product ion ms/ms 50–1800. Ions observed in the TOF-MS scan exceeding a threshold of 100 cps and a charge state of +2 to +5 were set to trigger the acquisition of product ion. The acquired data was processed using SCIEX OS software (AB SCIEX, Concord, Canada).

The LC-MS/MS data was imported to the PEAKS studio (Bioinformatics Solutions Inc., Waterloo, ON, Canada, version 7.0) with the assistance of MS Data Converter (Beta 1.3, http://sciex.com/software-downloads-x2110). Peptides were analyzed using PEAKS v7.0 (BSI, Canada) against the protein database built from the *T. pisana* transcriptome-derived protein data. PEAKS used the following parameters: (i) precursor ion mass tolerance, 0.1 Da; (ii) fragment ion mass tolerance, 0.1 Da (the error tolerance); (iii) tryptic enzyme specificity with two missed cleavages allowed; (iv) monoisotopic precursor mass and fragment ion mass; (v) a fixed modification of cysteine carbamidomethylation; and (vi) variable modifications including lysine acetylation, deamidation on asparagine and glutamine, oxidation of methionine and conversion of glutamic acid and glutamine to pyroglutamate. *De novo* sequencing of peptides, database search and characterizing specific PTMs were used to analyze the raw data; false discovery rate (FDR) was set to ≤ 1%, and [-10*log(p)] was calculated accordingly, where p is the probability that an observed match is a random event.

### Gland protein isolation, SDS-PAGE, and western blot

Albumen and mucous glands from reproductive stage *T. pisana* were excised and homogenized in a phosphate buffered saline (1xPBS; 100 mg/ml). Following centrifugation at 12,000 × g for 20 min at room temperature, the supernatant was pipetted into new tubes and protein concentration was determined using a Nanodrop 2000c at 280 nm. Sample buffer (2x Laemmli (Bio-Rad) with 2-mercaptoethanol (to 10% final concentration) was added to 30 µg protein and boiled at 90°C for 5 min. Protein separation was performed using SDS-PAGE with a MINI Protean precast Gel TGX (7.5%). Gel proteins were transferred to a 0.2 µm PVDF membrane (Transfer-Blot Turbo Transfer System Transfer Pack; Bio-Rad) using a Bio-Rad Transblot Transfer System cassette, set at 2.5A, 19 V for 3 min, then the membrane was removed, blot dried and stored at 4°C.

After washing the membrane in 1 x PBS, a blocking buffer of 4% skim milk in 1 x PBS was added and the membrane rocked for 2 h at room temperature. Zebrafish anti-AChE (PA5–117711; Invitrogen) polyclonal antibody was added at 1:1000 dilution in 1 x PBS, and incubated on a platform mixer overnight at 4°C. The membrane was washed 3 times for 5 min each in 1 × PBS. A secondary anti-IgG infrared 700 nm (Li-COR) was added at a dilution of 1:1000 in 1 × PBS and rocked for 2 h at room temperature. The membrane was washed 3 times in 1 x PBS for 2 min each, then dried and viewed at 700 nm on a Li-Cor Odyssey CLx Millenium Science imaging system.

### Mucous gland histology and immunofluorescence

Mucous glands from reproductive *T. pisana* were dissected and set in 70% fixative ethanol overnight at 4°C. For histology and immunofluorescence, samples were further dehydrated in 100% ethanol, then incubated in xylene and paraffin. Tissues were embedded, followed by sectioning at 10 µm using a ProSciTech computer microtome. Sections were deparaffinized in 2 changes of xylene for 10 min each, followed by rehydration. For histology, sections were stained with hematoxylin and eosin (Merck) following protocols previously described [40]. After staining, sections were cleared in xylene

 

twice for 5 min and mounted with Permount medium (ProSciTech). Images were taken using an upright Leica microscope with a CCD camera.

For immunofluorescence, unstained deparaffinized sections were incubated in a blocking solution containing 4% goat serum (Invitrogen) in 1 x PBS with 0.1% Tween (PBST) was added to the slides which were incubated at room temperature for 1 h. Then, zebrafish anti-AChE (PA5–117711; Invitrogen) polyclonal antibody was added at 1:500 dilution in PBST and incubated at 4°C overnight. Slides were washed 3 times in PBST for 5 min each, followed by incubation with secondary antibody (anti-rabbit IgG 488; Invitrogen) at 1:200 dilution for 1 h at room temperature. Slides were washed in PBST for 5 min and tissue sections counterstained with DAPI (Invitrogen) at 1 mg/ml for 5 min. The solution was discarded, and slides mounted with coverslips with antifade fluorescence mounting medium (Abcam). Slides were stored in the dark at 4°C, prior to microscopy. Images were taken using an upright Leica fluorescence microscope with a CCD camera.

### Snail mucus trail immunofluorescence

Reproductive and non-reproductive stage *T. pisana* were rinsed in water, then placed on poly-L-lysine coated slides (ProSciTech) to move across the slides. The slides were transferred to an incubator for 1 h at 45°C. Paraformaldehyde (4%), was pipetted onto the slides, left to sit for 1 h at room temperature, washed in 1 x PBS, and stored in 70% ethanol at 4°C until required. Immunofluorescence and imaging was performed as described in the previous section.

## Results

### Identification and phylogenetic analysis of Theba pisana AChE-like proteins

The *T. pisana* reference transcriptome database was searched for AChE-like proteins, resulting in the identification of 21 AChE-like proteins, which had the most similarity to a single *Candidula unifasciata* AChE-like protein (S1 Fig and S1 Table). The majority of *T. pisana* AChE-like proteins were deemed partial-length based on the absence of initiator methionine and/or absence of a stop codon. Those that were deemed full-length ranged from 139 to 557 amino acids in length. Phylogenetic analysis of *T. pisana* AChE-like proteins indicated three distinct clades were present (Fig 1A). Gene expression of *T. pisana* AChE-like genes was investigated utilising RNA-seq information obtained from the mucous gland, albumen gland, foot muscle and cerebral ganglia during reproductive and non-reproductive stages (GenBank accession number PRJNA858108) (Fig 1B). Most AChE-like proteins were relatively abundant in the mucous gland at reproductive stage, including those identified in the trail mucus (S2 Table). Others showed relatively high gene expression in the cerebral ganglia at both reproductive and non-reproductive stages.

### Protein identification from reproductive-stage trail mucus

Trail mucus proteins collected from reproductive stage *T. pisana* were fractionated using SDS-PAGE, then the major band at ~60 kDa (Fig 2A) was taken and processed through mass spectrometry. Four different AChE-like proteins were identified, all containing N-terminal signal peptides, multiple cysteine residues, and varied in size from 351–552 amino acids; the AChE-like protein containing the largest precursor protein is shown in Fig 2B. Trail mucus AChE-like proteins (contigs 5507, 7063, 7249, 4592) were highly expressed in the mucous glands of reproductive stage snails, and present in separate phylogenetic clades compared to the vertebrate AChE proteins (Fig 1A).

### Immunolocalisation of AChE-like proteins in mucous glands and trail mucus

A commercial antibody generated against zebrafish AChE (NP_571921) was used to localise AChE-like proteins in *T. pisana*. Initially, the specificity was tested using western blotting, showing an immunoreactive band in the mucous gland of expected size (~60 kDa) (Fig 3). The albumen gland demonstrated 2–3 immunoreactive bands below 25 kDa.

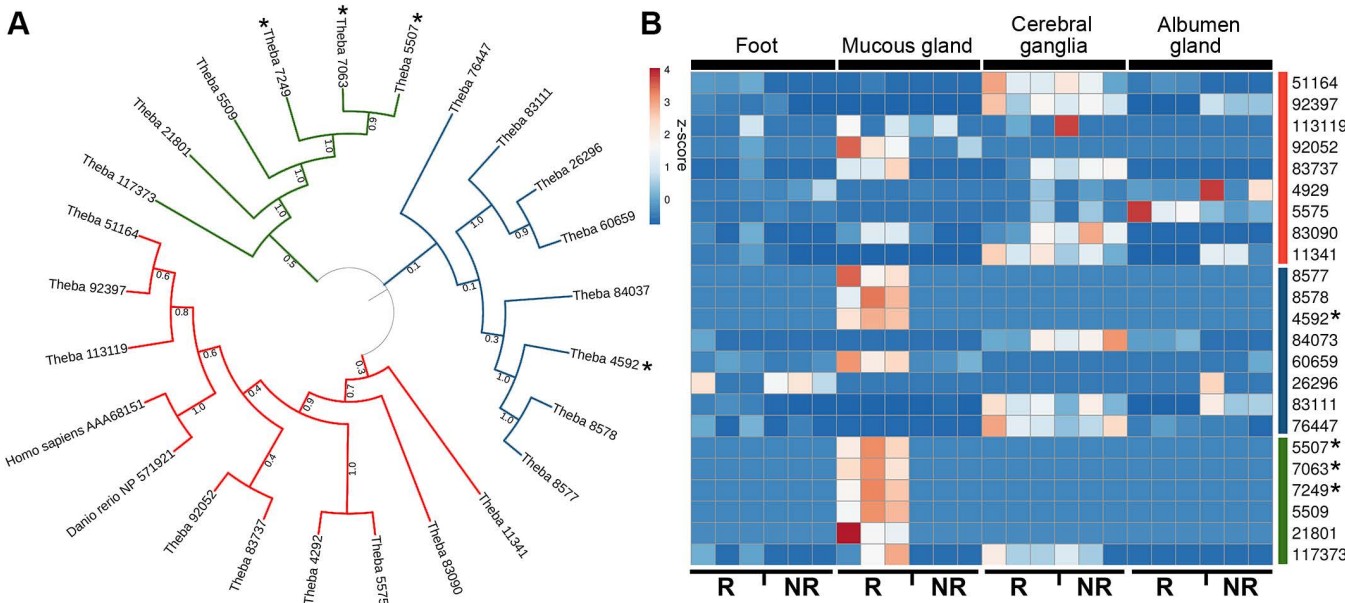

**Fig 1. Characterisation of AChE-like proteins genes in *Theba pisana*.** (A) Phylogenetic tree of AChE-like proteins. Three clades are represented in red, blue, and green. *, denotes AChE-like proteins found in reproductive *T. pisana* trail mucus. (B) Heat map of the AChE-like protein gene expression in 4 different tissues at reproductive (R) and non-reproductive (NR) stages. Sidebars (red, blue, and green) correlate with the phylogenetic tree clades. AChE-like proteins found in the reproductive trail mucus are denoted with an asterisk. All *T. pisana* protein sequences can be found in S1 Fig.

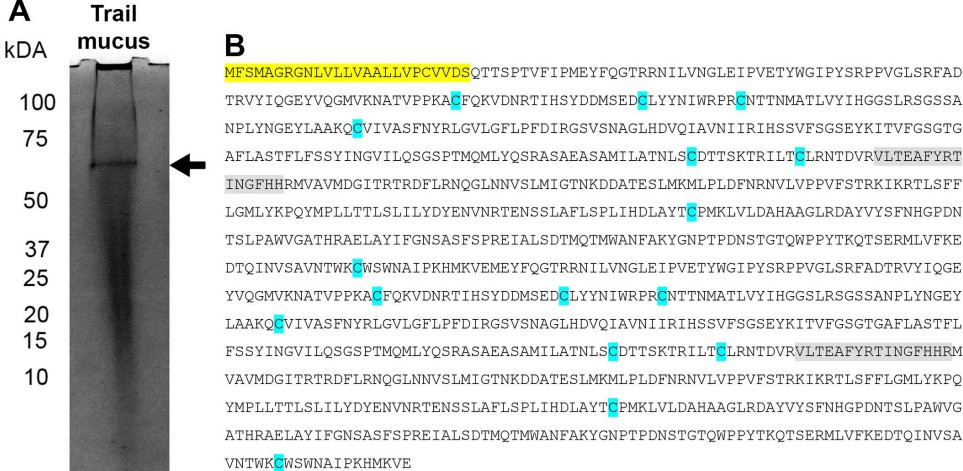

**Fig 2. Gel fractionation of reproductive stage *Theba pisana* trail mucus and protein identification.** (A) SDS-PAGE with Coomassie blue stain. The arrow indicates the protein band isolated for mass spectrometry analysis. (B) Precursor protein sequence for AChE-like protein (contig 5507), including signal peptide (yellow), cysteine residues (blue), and mass spectrometry peptide matches (grey).

Following confirmation of anti-AChE specificity, reproductive stage mucous gland tissue and trail mucus were investigated. Mucous gland tissue sections stained with hematoxylin and eosin demonstrated the general glandular cell organisation, highlighted by extensive epithelial folds containing secretory mucous gland cells, villi, and nuclei (Fig 4A, B). Immunofluorescent localisation revealed the presence of AChE-like protein associated with mucous gland cells (Fig 4C,

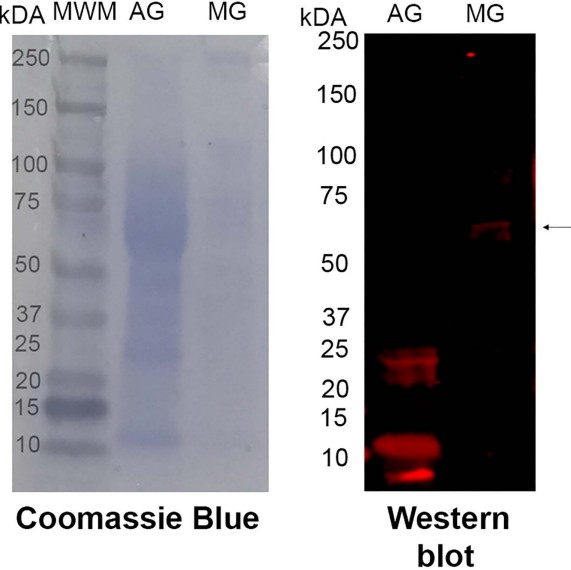

**Fig 3. Western blot image showing the presence of trail mucus AChE-like protein in the mucous and albumen gland (arrow) of *Theba pisana* at the reproductive stage.**

D). In trail mucus of reproductive stage *T. pisana* individuals, AChE-like protein was observed across linear lines, presumably mucous threads, but also scattered (Fig 4E–G). In contrast, a low level of immunoreactive material was found in non-reproductive stage *T. pisana* trail mucus (S2 Fig).

## Discussion

Experimental and theoretical rationales regarding the purpose of land snail trail mucus have been an important area of research for more than 40 years [41–44]. Obtaining knowledge as to why snails produce mucus for locomotion, adhesion, reproduction, aggregation, and various other functions has been integral to the elucidation of their ecological and evolutionary development. Despite this, the role of most proteins produced in trail mucus is relatively unknown. Here, we investigated a major protein component of trail mucus in the reproductive stage of *T. pisana*, compared with the non-reproductive stage, and propose a function for trail mucus associated with pesticide resistance.

There exists a vast amount of information surrounding vertebrate AChE due to its significant role in terminating acetylcholine-mediated neurotransmission [45–48]. In molluscs, this role has been confirmed through studies in *Aplysia* [49] and *Mytilus* [50]. However, AChE potential role in overcoming OPs has been more intensively studied in invertebrates, particularly in the fruit fly (*Drosophila*) [51] and ticks (*Rhipicephalus*) [52]. Molluscan gastropods are also excellent invertebrate models for investigating harmful biomarkers in an environment, especially since AChE activity functions in the space of inhibition against certain pesticides [53].

Towards a better understanding of land snail AChE-like proteins, the trail mucus from *T. pisana* was analyzed through mass spectrometry, which resulted in the identification of four AChE-like precursor proteins, all containing multiple cysteine residues. We identified that *T. pisana* may produce up to 21 AChE-like proteins based on tissue transcriptomic interrogation. Of them, we could report only eight as full-length sequences (which included the trail mucus AChE-like proteins), likely due to inefficient *de novo* assembly issues. Currently, there is no reference genome for *T. pisana*, which if available, would facilitate a more accurate and complete transcriptome assembly. The species with the highest sequence identity to *T. pisana* AChE-like proteins was *Candidula unifasciata*, a land snail with a slightly more aggressive nature for colonization

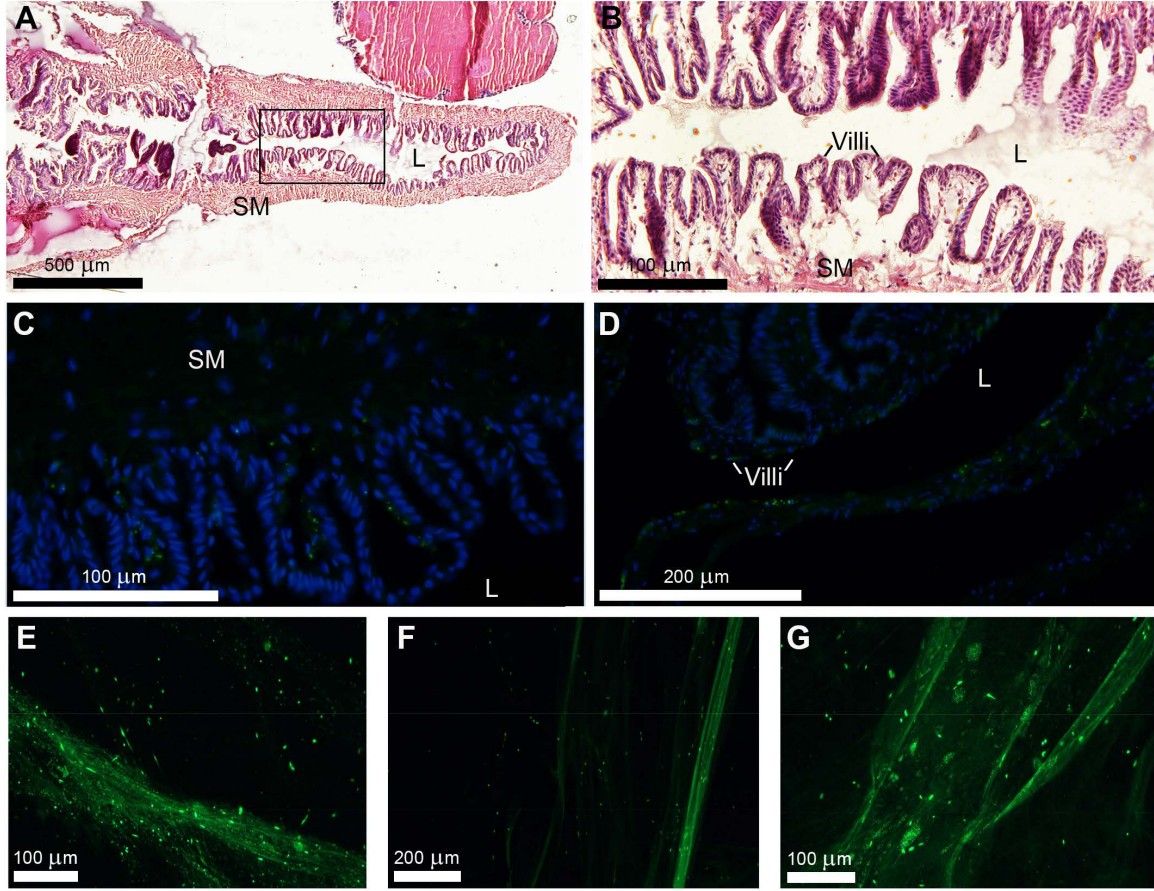

**Fig 4. Immunolocalization of AChE-like protein in *Theba pisana* mucous glands and reproductive stage trail mucus.** (A) Hematoxylin and eosin-stained section of mucous gland, and (B) magnified region shown in (A) by black box. (C, D). Immunolocalization of AChE-like protein (green) in mucous gland, and counterstained with DAPI (blue). (E, F, G) Immunolocalization of AChE-like protein (green) in reproductive stage trail mucus. L, lumen; SM, smooth muscle.

in regions of Southern France and Germany [54]; although both *Candidula* and *Theba* have a high threshold for heat tolerance [55,56]. Of interest is *C. unifasciata,* which appears to contain only one AChE-like protein, based on genome annotation [54]. Thus, the expansion of AChE-like proteins could be specific to *T. pisana*; alternatively, more information is needed at the transcriptomic and proteomic levels for *C. unifasciata* to accurately know the AChE-like protein number.

It was surprising to observe large amounts of AChE-like proteins in *T. pisana* trail mucus, especially given its recognized role in neural signalling. AChE is well-known as a primary neural cholinesterase that breaks down acetylcholine and choline esters, which function as neurotransmitters [57,58]. In agricultural pest control, OPs bind to AChE, which leads to a reduction in the hydrolysis of acetylcholine and therefore creates an accumulation of acetylcholine in the neural synaptic cleft [59]. Overstimulation of acetylcholine in postsynaptic membranes, left continuously open, destroys cholinergic transmission which severs communication in the nervous system, blocking nerve conduction and therefore achieves the desired effect of control [59–61]. Thus, our discovery of trail mucus AChE-like proteins deserved more in-depth investigation.

We found that *T. pisana* AChE-like proteins were phylogenetically distributed into three clades. Of interest, the vertebrate AChE proteins included in the phylogeny (*Homo sapiens* and *Danio rerio*), appeared in a separate clade from the

four-trail mucus AChE-like proteins, yet similar to nine *T. pisana* AChE-like proteins with relatively high neuronal gene expression. We speculate that these may function as conventional AChE, facilitating breakdown or neural acetylcholine. As previously mentioned, *T. pisana* are most active during the reproductive stage and thus are more vulnerable to pesticides. This may explain the heightened requirement for a protective mechanism to avoid the toxicity of pesticide residues [62–68]. For pesticides to be functional and effective, it is theorized that organophosphate and carbamate molecules must successfully penetrate amino acid residues that line neural cell walls before binding an active serine residue [69,70]. *In vitro* studies on lepidopteran larvae found that conformational changes in the structure of AChE have allowed invertebrates to express more than one AChE gene when exposed to stress, harmful conditions or hypoxia [71].

In this study, trail mucus analysis was performed on reproductively active adult snails. At the reproductive stage, snails are generally at their most active [13,72]; therefore, trail mucus was relatively easy to obtain in high yields. Targeting the most abundant protein based on SDS-PAGE with Coomassie staining, 4 AChE-like proteins were identified. Comparative RNA-seq analysis found that gene expression of these proteins was exclusive to the mucous glands of reproductive stage individuals. AChE-like proteins have never been reported in land snail mucus, even following a recent thorough investigation of the common garden snail, *Cornu aspersum* [37], a notorious garden pest also known for invasive tendencies and crop infestation [73–75]. In that mucomic investigation, it was found that there were three types of mucus secretions, involved with shielding, adhesion, and lubrication [73].

A commercially available anti-zebrafish AChE (full sequence) was used to localize AChE-like proteins in *T. pisana* mucous gland tissue and trail mucus. Before localization, specificity was validated through western blotting, specifically a ~60 kDa protein band in the mucous tissue extract, which aligns with the expected size of the trail mucus AChE-like proteins. Spatial immunolocalizations within the mucous gland tissue confirmed that production and secretion most likely occurred in the mucous gland secretory cells, from which it could progress into the snail's trail mucus. Immunolocalization of trail mucus demonstrated AChE-like protein distribution throughout the mucus threads. This finding aligns with other invertebrate AChE-like protein research, which shows the versatility of secretory AChE functions, specifically towards resilience against OP target sites [57,76–79]. Moreover, sequestered non-neuronal AChE proteins have been implicated in the inactivation of pathogenic or predatory invasion, and help regulate physiological functions [57,80–82].

A recent study of another major pest snail, *C. virgata*, found that during reproductive stage (and compared to non-reproductive stage), the trail mucus contained a relative abundance of various proteins, including AChE, as well as leukocyte elastase inhibitors, cathepsin and agglutinin, all of which are known to have roles in immunity, defence and cell signalling [38]. That study was focussed on total protein profile changes between reproductive stages, which likely inferred reproductive readiness [38]. However, it does open up the possibility that other genes and proteins are of importance in pesticide resistance.

There is emerging evidence that interdisciplinary investigation in pest management, including molecular-based research on mucus, can better assist in developing practical agricultural applications [37,83]. Numerous tests using baits composed of OP targeted towards AChE activity have resulted in successful immediate outcomes; however, enzymatic targets open the door to species adaptation and subsequent resistance. Despite many agricultural attempts at control towards mitigation, these toxic pesticide methods are at best temporary measures without assurance of longevity. Risks associated with using conventional pesticides have been identified to suppress communities of land and water-dwelling snail pests, yet this has resulted in secondary outbreaks and susceptibility to a vicious cycle of pollution [84,85]. The knowledge obtained from our research at the molecular level could lead to further research into countering toxin-associated resistance. Alternatively, it may highlight the need for more natural alternatives to commercial pesticides, as has been developed through research into insect pheromones.

## Conclusions and future research

In summary, this research provides new information regarding the makeup of snail trail mucus, using a significant invasive pest snail, *T. pisana*. We found that *T. pisana* secretes AChE-like proteins into trail mucus during the reproductive stage,

a time in which they are most active and vulnerable to pesticides. Based on a multidisciplinary investigation, we demonstrated that AChE-like proteins are produced in the snail mucous gland and ultimately spread into the trail mucus, a likely mechanism to protect the snail from the direct impact of organophosphates. Although acetylcholinesterase had been speculated to be a mechanism by which insects have overcome pesticides, this is the first description of this function in a land snail.

For future research, it would be of interest to perform biochemical analyses focused on the bioactivity of trail mucus AChE-like proteins. This would assist in determining whether the function of AChE-like proteins in land snails includes both a neuronal role along with the ability to scavenge organophosphates. *Theba pisana* has diversified the number and location of AChE-like proteins, compared to vertebrates. It would also be of interest to observe any differences in *T. pisana* that have never (or rarely) been exposed to pesticides, if any such populations can be found. This would lead to a better understanding of the molecular mechanism of variation. Finally, in recognition of the ability of snails to adapt and overcome toxin-mediated biocontrol measures, the potential for pheromones, likely also present in trail mucus, should be further investigated to overcome current and emerging land snail pest invasions.

## Supporting information

**S1 Fig.  Acetylcholinesterase-like protein sequences found in *Theba pisana*.**
(DOCX)

**S2 Fig.  Immunolocalization of AChE-like protein in non-reproductive *Theba pisana* trail mucus.** (A, B) Immunolocalization showing low/no acetylcholinesterase-like protein (green) in trail mucus.
(DOCX)

**S1 Table.  Summary of *Theba pisana* AChE-like proteins (with *Homo sapien* and *Danio rerio*), including gene ID, best species match, length, pI and molecular weight.**
(DOCX)

**S2 Table.  Summary of *Theba pisana* reproductive trail mucus AChE-like protein gene expression in mucous gland, albumen gland, foot and cerebral ganglia tissue.** Expression is in transcripts per million. R, reproductive stage; NR, non-reproductive stage.
(DOCX)

## Acknowledgments

Thanks to Stuart J. Smith and Paul A. Wright for lab assistance and computer support.

## Author contributions

**Conceptualization:** Inaliguyau R. T. Lutschini, Kate R. Ballard, Scott F. Cummins.

**Data curation:** Inaliguyau R. T. Lutschini.

**Formal analysis:** Inaliguyau R. T. Lutschini, Tianfang Wang, Scott F. Cummins.

**Investigation:** Inaliguyau R. T. Lutschini, Scott F. Cummins.

**Methodology:** Inaliguyau R. T. Lutschini, Tianfang Wang, Scott F. Cummins.

**Supervision:** Scott F. Cummins.

**Validation:** Inaliguyau R. T. Lutschini.

**Visualization:** Tianfang Wang, Scott F. Cummins.

**Writing – original draft:** Inaliguyau R. T. Lutschini, Scott F. Cummins.

**Writing – review & editing:** Kate R. Ballard, Tianfang Wang, Scott F. Cummins.

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
