## [Decision Letter · Decision Letter 0]

14 Mar 2025

PONE-D-25-02235Acetylcholinesterase-like proteins are a major component of reproductive trail mucus in the invasive pest land snail, Theba pisanaPLOS ONE

Dear Dr. Cummins,

Thank you for submitting your manuscript to PLOS ONE. After careful consideration, we feel that it has merit but does not fully meet PLOS ONE’s publication criteria as it currently stands. Therefore, we invite you to submit a revised version of the manuscript that addresses the points raised during the review process.

We look forward to receiving your revised manuscript.

Kind regards,

Ebrahim Shokoohi

Academic Editor

PLOS ONE

Additional Editor Comments:

Dear Authors

Now, we have received the feedbacks from the Referees, and you should address them point by point. The comments are given for your reference.

Reviewers' comments:

Reviewer's Responses to Questions

**Comments to the Author**

1. Is the manuscript technically sound, and do the data support the conclusions?

Reviewer #1: Yes

Reviewer #2: Yes

2. Has the statistical analysis been performed appropriately and rigorously? 

Reviewer #1: Yes

Reviewer #2: Yes

3. Have the authors made all data underlying the findings in their manuscript fully available?

Reviewer #1: Yes

Reviewer #2: Yes

4. Is the manuscript presented in an intelligible fashion and written in standard English?

Reviewer #1: Yes

Reviewer #2: Yes

5. Review Comments to the Author

Reviewer #1: The manuscript described the Acetylcholinesterase-like proteins are a major component of reproductive trail mucus in the invasive pest land snail, Theba pisana. The article is well-written but it is not acceptable for publication in the current format, however, it is acceptable after a minor revision. The following issues should be addressed by the authors.

Specific comments:

1. The author needs to provide the reference for following the specific protocol for the Identification and characterisation of acetylcholinesterase-like genes (Line 141-160).

2. Italicize the word In Vitro (Line 367).

3. Kindly add full stop before the beginning of a new sentence i.e., For pesticides in the Line 364.

4. It’s advisable to authors to strictly follow one font style throughout the manuscript.

5. Provide single space before the ‘a notorious garden pest’.

6. Locate the epithelial folds and other described things in the Figure 4 probably using arrows.

7. Author needs to mention the accession number of AChE-like proteins genes in Theba pisana, in the Figure 1A and in the text also.

8. Kindly go through the manuscript for proof reading again to avoid text mistakes.

9. What is the hypothesis behind this study?

10. Have authors looked for the possibility of other genes other than acetylcholinesterase-like genes in the trail mucous of snails and on the idea that other genes might be involved in the resistance against the pesticides?

Reviewer #2: 1- Please explain more details about Acetylcholinesterase analysis in material and method

2- Please improve the English style

3- What is the effect of Acetylcholinesterase in the life cycle of pest land snail

6. PLOS authors have the option to publish the peer review history of their article (what does this mean? ). If published, this will include your full peer review and any attached files.

**Do you want your identity to be public for this peer review?** For information about this choice, including consent withdrawal, please see our Privacy Policy .

Reviewer #1: No

Reviewer #2: No

---

## [Author Response · Author response to Decision Letter 1]

27 Mar 2025

Reviewer #1:

Specific comments:

1. The author needs to provide the reference for following the specific protocol for the Identification and characterisation of acetylcholinesterase-like genes (Line 141-160).

RESPONSE: A reference has now been included to the Methods that justifies the protocol used – i.e. The general protocol used to identify and characterise snail genes was as described by Ballard et al., 2021 and Ballard et al., 2025.

2. Italicize the word In Vitro (Line 367).

RESPONSE: In vitro has now been italicized.

3. Kindly add full stop before the beginning of a new sentence i.e., For pesticides in the Line 364.

RESPONSE: A full stop has been added.

4. It’s advisable to authors to strictly follow one font style throughout the manuscript.

RESPONSE: We agree and have ensure consistency in font style throughout. i.e. Times New Roman, font size 12.

5. Provide single space before the ‘a notorious garden pest’ (line 378).

RESPONSE: A single space has now been added.

6. Locate the epithelial folds and other described things in the Figure 4 probably using arrows.

RESPONSE: To assist in recognition of the relevant histological features, we have annotated the images to provide labelled spatial location of the mucous gland lumen, smooth muscle and villi that represent the epithelial folds. In addition, the scale bars have been updated for clarity.

7. Author needs to mention the accession number of AChE-like proteins genes in Theba pisana, in the Figure 1A and in the text also.

RESPONSE: We now include in the main text (as S1 File and S2 File) and Figure 1 legend (as S1 File) where all AChE-like sequences used in the analyses can be found. The NCBI BioProject database under accession number PRJNA858108 is provided in the Methods and Resources for those who want to assess the raw data.

8. Kindly go through the manuscript for proof reading again to avoid text mistakes.

RESPONSE: All authors have proofread the manuscript prior to our resubmission.

9. The following edits have been made as suggested:

Line 250 – edited Microscope images were taken using an upright Leica fluorescence microscope with a CCD camera, to ‘Images were taken using an upright Leica fluorescence microscope with a CCD camera.’

Line 256– edited the 4% paraformaldehyde was pipetted onto each slide… to ‘Paraformaldehyde (4%) was pipetted onto each slide’…

Line 271 – edited the majority of AChE-like proteins were relatively... to ‘Most AChE-like proteins were’...

Line 302 – edited Initially, specificity was tested using western blot… to ‘Initially, the specificity was tested using western blot’…

Line 386 – edited 4 AChE-like proteins were comparative identified… to ‘4 AChE-like proteins were identified’…

Line 401 – edited which shows the versatility of secretory AChE function to ‘which shows the versatility of secretory AChE functions…’

Line 419 – edited communities of land and water dwelling snail pests, however, to ‘communities of land and water dwelling snail pests, yet…’

Line 431 – edited and ultimately get spread into the… to ‘and ultimately spread into the’ …

Line 436 to 445– edited This would assist in determining its ability to function similarly to its natural neuronal role, or purely to scavenge organophosphates. It is clear that T. pisana has diversified the number and location of AChE-like proteins, compared to vertebrates, but it would be of interest of observe… to

‘This would assist in determining whether the function of AChE-like proteins in land snails includes both a neuronal role along with the ability to scavenge organophosphates. Theba pisana has diversified the number and location of AChE-like proteins, compared to vertebrates. It would be of interest to observe’…

9. What is the hypothesis behind this study?

RESPONSE: We thank the reviewer for recognising no clear hypothesis statement. In the Introduction section, we have added the study hypothesis - We hypothesized that land snails secrete AChE proteins into their trail mucus, which may function to neutralize pesticides.

10. Have authors looked for the possibility of other genes other than acetylcholinesterase-like genes in the trail mucous of snails and on the idea that other genes might be involved in the resistance against the pesticides?

RESPONSE: We agree, and following the recently published manuscript in Biology - https://www.mdpi.com/2079-7737/14/3/294 - where the trail mucus of a related land snail species (Cernuella virgata) was investigated, we have added to the discussion information that describes multiple different proteins that are temporally present in adult C. virgata mucus. We postulate that there are other proteins that could help facilitate pesticide resistance and should be further investigated.

Reviewer #2:

1- Please explain more details about Acetylcholinesterase analysis in material and method

RESPONSE: To help clarify details of the AChE identification and analysis, we have added new information to the Material and Method section:

- Line 143 – The general protocol used to identify and characterise snail genes was as described by Ballard et al., 2021 (1, 2)

- Line 165 – Trail mucus was collected using a method previously described by Ballard et al., 2021(1, 2)

2- Please improve the English style

RESPONSE: All authors have proofread the manuscript prior to our resubmission. Changes include those listed below, with line numbers based on original submission -

Line 13 – edited yet are often broad-spectrum and species develop resistance… to ‘yet are often broad-spectrum, leading to the development of resistance in target species’…

Line 19 – edited clades; one clade included… to ‘clades, with clade including’…

Line 52 – edited (e.g., wheat, barley, oat), which can clog farming machinery… to ‘(e.g., wheat, barley, oat); as a result, snails can be inadvertently harvested which can clog farming machinery’…

Line 53 – edited Aestivation is a behavioural response to higher temperature that… to ‘Aestivation is a behavioural response to higher temperatures that’…

Line 85 – edited This research investigated a potential role for resistance-modified AChE in invasive pest land snails, using T. pisana. Using tissue- and … to ‘We hypothesized that T. pisana secrete AChE proteins into their trail mucus, which may function to neutralize pesticides. Utilising tissue- and’…

Line 89 – edited This finding helps provide some explanation as to the potential molecular cause of pesticide resistance… to ‘This finding helps to provide some explanation as to the potential of molecular signalling mechanisms involved in pesticide resistance and’…

Line 128 – edited immediately snapped frozen… to ‘immediately snap-frozen’…

Line 166 – edited T. pisana were, rinsed… to T. pisana were rinsed’…

Line 197 – edited the LC-MS/MS data were imported… to ‘the LC-MS/MS data was imported’…

Line 214 – edited into new tubes, then… to ‘into new tubes and;’…

Line 223 – edited was added and rocked… to ‘was added and the membrane rocked’…

Line 233 – edited samples were further dehydrated to 100% ethanol… to ‘samples were further dehydrated in 100% ethanol’…

Line 242 – edited in 1 x PBS, 0.1% Tween (PBST)… to ‘in 1 x PBS, then 0.1% Tween (PBST)’…

Line 243 – edited added to slides and incubated… to ‘added to the slides which were incubated’…

Line 256 – edited onto each slide and left for 1 h… to ‘onto the slides, covered, and left to sit for 1 h’…

Line 270 – edited addition of (GenBank accession number PRJNA858108)’…

Line 386 - edited 4 AChE-like proteins were comparative identified; comparative RNA-seq… to ‘4 AChE-like proteins were identified; Comparative RNA-seq’…

Line 388 – edited AChE-like proteins had never… to ‘AChE-like proteins have never’…

Line 416 – edited agricultural gains to mitigate invasiveness… to ‘agricultural attempts of control towards mitigation’…

Line 421 – edited could lead to research… to ‘could lead to further research’…

Line 427 – edited this research provided… to ‘this research provides’…

Line 432 – edited likely mechanism to avoid the direct impact… to ‘likely mechanism to protect the snail from the direct impact’…

Line 433 – edited speculated as a mechanism… to ‘speculated to be a mechanism’…

Line 434 – edited first description of a land snail… to ‘first description of this function in a land snail’…

3- What is the effect of Acetylcholinesterase in the life cycle of pest land snail

RESPONSE: In the discussion section, we have added information regarding the recognised effect of acetylcholinesterase on the life cycle of the snail, and the molluscs in general:

There exists a vast amount of information surrounding vertebrate AChE due to its significant role in terminating acetylcholinesterase-mediated neurotransmission (3-6). In molluscs, this role has been confirmed through studies in Aplysia (7) and Mytilus (8). However, AChE potential role in overcoming OPs has been more intensively studied in invertebrates, particularly in the fruit fly (Drosophila) (9) and ticks (Rhipicephalus) (10). Molluscan gastropods are also excellent invertebrate models for investigating harmful biomarkers in an environment, especially since AChE activity functions in the space of inhibition against certain pesticides (11).

References

1. Ballard KR, Klein AH, Hayes RA, Wang T, Cummins SF. The protein and volatile components of trail mucus in the Common Garden Snail, Cornu aspersum. PloS one. 2021;16(5):e0251565-e.

2. Ballard KR, Ventura T, Wang T, Elizur A, Cummins SF. Mucus Trail Proteins May Infer Reproductive Readiness for Land Snails. Biology. 2025;14(3):294.

3. Herz F, Kaplan E. A review: human erythrocyte acetylcholinesterase. Pediatric Research. 1973;7(4):204-14.

4. Patočka J, Kuča K, Jun D. Acetylcholinesterase and butyrylcholinesterase–important enzymes of human body. Acta Medica (Hradec Kralove). 2004;47(4):215-28.

5. Saldanha C. Human erythrocyte acetylcholinesterase in health and disease. Molecules. 2017;22(9):1499.

6. Thomsen T, Kaden B, Fischer J, Bickel U, Barz H, Gusztony G, et al. Inhibition of acetylcholinesterase activity in human brain tissue and erythrocytes by galanthamine, physostigmine and tacrine. 1991.

7. Srivatsan M, Peretz B. Acetylcholinesterase promotes regeneration of neurites in cultured adult neurons of Aplysia. Neuroscience. 1997;77(3):921-31.

8. Santos GPCd, Assis CRDd, Oliveira VM, Cahu TB, Silva VL, Santos JF, et al. Acetylcholinesterase from the charru mussel Mytella charruana: kinetic characterization, physicochemical properties and potential as in vitro biomarker in environmental monitoring of mollusk extraction areas. Comparative Biochemistry and Physiology Part C: Toxicology & Pharmacology. 2022;252:109225.

9. Fournier D, Mutero A, Pralavorio M, Bride J-M. Drosophila acetylcholinesterase: Mechanisms of resistance to organophosphates. Chemico-Biological Interactions. 1993;87(1):233-8.

10. Singh H, Nandi A, Singh NK. STATUS OF ACETYLCHOLINESTERASE MEDIATED ORGANOPHOSPHATE RESISTANCE IN CATTLE TICK, RHIPICEPHALUS MICROPLUS (ACARI: IXODIDAE). Exploratory Animal & Medical Research. 2024;14.

11. da Silva Lucas RM, Gonçalves FJ, Rodrigues FA, de Oliveira LCC. Acetylcholinesterase Activity Evaluated in Molluscs–A Bibliometric Analysis. Brazilian Journal of Development. 2025;11(1):e76398-e.

---

## [Editor Report · Decision Letter 1]

8 Apr 2025

Acetylcholinesterase-like proteins are a major component of reproductive trail mucus in the invasive pest land snail, Theba pisana

PONE-D-25-02235R1

Dear Dr. Scott F Cummins,

We’re pleased to inform you that your manuscript has been judged scientifically suitable for publication and will be formally accepted for publication once it meets all outstanding technical requirements.

Kind regards,

Ebrahim Shokoohi

Academic Editor

PLOS ONE

Additional Editor Comments (optional):

All concerns has been addressed.
---

## [Editor Report · Acceptance letter]

PONE-D-25-02235R1

PLOS ONE

Dear Dr. Cummins,

I'm pleased to inform you that your manuscript has been deemed suitable for publication in PLOS ONE. Congratulations! Your manuscript is now being handed over to our production team.

Kind regards,

on behalf of

Dr. Ebrahim Shokoohi

Academic Editor

PLOS ONE